# Lipid Mediators Metabolic Chaos of Asthmatic Mice Reversed by Rosmarinic Acid

**DOI:** 10.3390/molecules28093827

**Published:** 2023-04-30

**Authors:** Tuo Qin, Xiaojuan Rong, Xiaohui Zhang, Lingfei Kong, Yutong Kang, Xuanlin Liu, Mengying Hu, Handong Liang, Cai Tie

**Affiliations:** 1State Key Laboratory Coal Resources and Safe Mining, China University of Mining and Technology-Beijing, Ding11 Xueyuan Road, Beijing 100083, China; 2School of Chemical and Environmental Engineering, China University of Mining and Technology-Beijing, Ding11 Xueyuan Road, Beijing 100083, China; 3Xinjiang Institute of Material Medica, South Xinhua Road 140, Urumqi 830004, China; 4State Key Laboratory of Natural and Biomimetic Drugs, Peking University, 38 Xueyuan Road, Beijing 100191, China

**Keywords:** asthma, lipid mediators (LMs), rosmarinic acid (RosA), inflammation

## Abstract

Background and objective: Asthma is a common chronic inflammatory disease of the airways with no known cure. Lipid mediators (LMs) are a kind of inflammatory signaling molecules which are believed to be involved in the development of asthma. *Hyssopus cuspidatus* Boriss. is a traditional Uyghur medicine, which is widely used in the treatment of asthma and other respiratory diseases. Extraction of *Hyssopus cuspidatus* Boriss. was reported to neutralize asthma symptoms. The purpose of the study was to investigate both the anti-inflammatory and immunoregulation properties of the *Hyssopus cuspidatus* Boriss. extract (SXCF) and its main active constituent, rosmarinic acid (RosA), in vivo. The effect of RosA, a major constituent of SXCF, was evaluated on an asthmatic model, with both anti-inflammatory and immunoregulation properties. Materials and methods: Anti-inflammatory effect of SXCF and RosA was assessed using OVA-induced asthma model mice by UPLC-MS/MS method. Results: Overall, RosA played a critical role in anti-asthma treatment. In total, 90% of LMs species that were significantly regulated by SXCF were covered. On the most important LMs associated with asthma, RosA equivalent induced similar effects as SXCF did. It is believed that some constituents in SXCF could neutralize RosA excessive impacts on LMs.

## 1. Introduction

Bronchial asthma is a chronic inflammatory airway disease, with typical clinical manifestations of recurring symptoms (including cough, wheezing, and chest tightness). Asthma is a chronic inflammation influenced by a variety of cells [1]. Currently, approximately 300 million people suffer from asthma worldwide [2]. Widely accepted treatments for asthma are inhaled bronchodilators and glucocorticoids [3]. A significant portion of asthma patients is well-controlled. The medicines help asthma patients control symptoms but cannot cure the disease. What is worse, 5% to 10% of asthma patients remain poorly controlled [4]. Both pharmaceutical academics and industry professionals are devoted to developing novel medicine to treat asthma. For those people who had been living in areas with a high prevalence of asthma for centuries, effective treatment options were developed, which led to a clue for the discovery of a new medicine. Xinjiang province is a historical asthma high prevalence area due to the desert climate. Uyghur medicine developed distinctive therapy solutions for asthma based on hundreds of years of treatment experience. *Hyssopus cuspidatus* Boriss. has been used to relieve cough, asthma and dispelling dampness for a long time in Uighur medicine. Many studies have proven the efficacy of *Hyssopus cuspidatus* Boriss. in treatment of Bronchial asthma [5,6]. Phenolic acid Rosmarinic acid (RosA) was determined to be one of the prime components in *Hyssopus cuspidatus* Boriss, with apparent anti-inflammatory effects in asthma [7,8]. RosA has been shown to have several potential therapeutic applications, including as an anti-allergic, anti-inflammatory, and antimicrobial agent [9]. It has also been studied for its potential to help with allergies, asthma, and other respiratory conditions. RosA works by inhibiting the production and activity of proinflammatory cytokines and enzymes in the body [10]. These cytokines and enzymes are responsible for triggering and sustaining the inflammatory response. It was reported that RosA protects the lung by inhibiting the cyclooxygenase (COX) and lipoxygenase (LOX) activities and complement activation [11,12]. Due to the limited knowledge on SXCF and RosA mechanism, the development of *Hyssopus cuspidatus* Boriss. was delayed [13].

LMs are potent signaling molecules that regulate multiple cellular responses through receptor-mediated pathways, including cell growth and death as well as inflammation/infection. Asthma is characterized by chronic inflammation of the airways in which there is an overabundance of eosinophils, mast cells, and activated T helper lymphocytes [14]. These inflammatory cells release mediators that then trigger bronchoconstriction, mucus secretion, and remodeling [14]. Th2 cytokines typically induce inflammation in asthma by promoting the production of additional inflammatory mediators, other cytokines and chemokines [15]. Torres. R et al. reported that Prostaglandin E2(PGE_2_) inhibited mast cell activity by activating the E-prostanoid 2 (EP2) receptor by regulating the PGE_2_-EP2-mast cell axis [16]. Leukotrienes (LTs) are a type of powerful proinflammatory molecules; they can cause bronchial smooth muscle contraction, increase vascular permeability, airway mucus secretion, chemotactic inflammatory cell infiltration, airway remodeling, etc. [17]. A large number of studies have shown that levels of LTs in asthma patients are higher than those of normal patients, whether in disease onset or stable stage [18]. The imbalance of LMs is associated with partial lesions of the airway, blood vessels and lung parenchyma.

After systemic LMs profiling containing Linoleic acid, arachidonic acid-related LMs products were produced. This revealed the differential impacts on the dysregulation of asthma modeling between SXCF and RosA. RosA was believed to be a mainly effective constituent of SXCF. However, some side reaction effects would be induced by RosA, which could be neutralized by other constituents in SXCF. It was believed that some constituents of SXCF neutralized the side reaction caused by RosA while maintaining the effects.

## 2. Results

### 2.1. Enrichment Analysis

In order to analyze 68 intersection targets between RosA target and asthma-related targets, GO and KEGG enrichment analyses were carried out. The top 10 significant biological processes (BP), cellular compositions (CC), and molecular functions (MF) are displayed in Figure 1B. BPs are mainly related to response to oxidative stress, cellular response to chemical stress, etc. CCs are mainly related to membrane raft, membrane micro domain, etc. MFs are mainly related to endopeptidase activity, metallopeptidase activity, etc. KEGG pathway analyses were performed to investigate the mechanism of RosA in treating asthma, and the top 30 significant pathways are shown in Figure 1A. It was determined that the pathways (removing the pathways related to cancer and those not related to asthma) were mainly enriched in the inflammatory factor pathway (IL-17 signaling pathway, TNF signaling pathway), arachidonic acid metabolism pathway, and so on.

### 2.2. Animal Ethology

After OVA sensitizing, the mice were observed for sneezing, scratching, and wheezing and scored according to the following scoring method. No sneezing: 0 point; sneezing < 4 times: 1 point; sneezing 4–10 times: 2 points; sneezing > 11 times: 3 points. No scratching: 0 point; mild scratching: 1 point; moderate scratching: 2 points; severe scratching: 3 points. No wheezing: 0 points, mild wheezing: 1 point; moderate wheezing: 2 points; severe wheezing: 3 points. Animal ethology of different groups were shown in Figure 2.

### 2.3. Histology and Serology

Compared to control mice, the bronchi and blood vessels of asthma mice were in-filtrated by a large number of inflammatory cells with mucosal epithelial degeneration and mucosal thickening (Figure 3A–D). With RosA and SXCF treatment, inflammatory cell infiltration, bronchial epithelial degeneration, and mucosal thickening were significantly inhibited.

Based on the histological changes observed in this experiment, the severity of pulmonary injury was graded using the following main indicators: peribronchial and perivascular inflammatory cell infiltration, bronchial obstruction, and interstitial inflammatory cell infiltration (Table 1).

Immunoglobulin E (IgE) is a key immunoglobulin in the pathogenesis of IgE-mediated related allergic diseases. As shown in Figure 3E, the serum IgE of the model group was almost twice that of the control group with *p* value < 0.05, which proves that the model group was successfully established.

### 2.4. Regulated Cytokine Levels of Target Organ

The levels of inflammatory and immune-related cytokines in the BALF of mice in each group were measured. As shown in Figure 4, a significant increase in TNF-α was observed (*p* < 0.05). TNF-α is a cytokine with various biological functions. It is mainly produced by activated monocytes, macrophages and lymphocytes, and participates in various inflammatory reactions. TNF-α can induce inflammation and immune cell infiltration, increase vascular permeability [19], and promote the expression of adhesion molecules in vascular endothelial cells [20]. TNF-α is one of the most important inflammatory cytokines which is produced by macrophages and mast cells through the IgE-mediated pathway and also induces Th2 cells to produce IL-4 [21].

IL-4 is a cytokine with various biological functions. IL-4 plays an important role in the activation and differentiation of CD4+ T cells. It stimulates the development of Th2 cells, which produce specific cytokines and participate in the humoral immune response. At the same time, it can also inhibit the development of Th1 cells, which participate in cellular immune responses [22,23,24]. Our result showed that IL-4 level nearly doubled in the model group, which means that the model was established successfully.

The IFN-γ level was significantly increased in OVA-induced mice compared to that of the control group. In the intervention groups, IFN-γ levels have a significant downregulation. It should be observed that IFN-γ level in both intervention groups was lower than that in control group. IFN-γ plays a complex role in regulating inflammation. While it can stimulate the production of pro-inflammatory cytokines, such as TNF-α and IL-1β, it can also inhibit the production of IL-4, IL-5, and IL-13, which are involved in the development of allergic and eosinophilic inflammation [25,26].

### 2.5. Characterization of Target Organ LMs Metabolism Network

The impacts on the LMs metabolism network in lung tissue and serum induced by modeling and medical intervention were obtained with PCA analysis. As shown in Figure 5A, the LMs metabolism network in the lung was restored partially both with SXCF and RosA. As shown in Figure 5B, similar regulation in serum was observed with SXCF treatment. However, RosA caused excessive regulation in serum.

As shown in Figure 5C,D, the heatmap validated the PCA results. In lung tissue, LMs decreased close to the control group both with SXCF and RosA treatment. In serum, limited recovery to the control group was obtained.

## 3. Discussion

Asthma can affect people of all ages and there is no cure for it. Therefore, suppressing inflammation by inhibiting inflammatory mediators is a reasonable treatment for asthma.

Natural products are increasingly recognized as valuable sources of pharmacotherapeutic agents for treating chronic diseases or for their biological activity. RosA is a phenolic compound found in plants and possessing anti-inflammatory properties [9]. RosA has been determined to significantly reduce airway inflammation, decrease the levels of Th1 cytokines and Th2 cytokines, and inhibit the activation of signaling pathways associated with inflammation, such as MAPK and NF-κB [27]. RosA has also been shown to reduce the infiltration of inflammatory cells into the airways, such as eosinophils and neutrophils, which are commonly found in the lungs of individuals with asthma [28].

Early research has shown a strong correlation between Th2 cytokine expression and allergic asthma. Regulation of Th2 cytokines is essential for the development of common allergic responses such as asthma. These cytokines trigger inflammatory responses, which can increase IgE secretion and inflammatory cell recruitment [29]. These symptoms are the primary pathophysiological indicators of allergic airway conditions. IL-4 has shown the effect of inhibiting the production of Th1 cells that secrete IFN-γ [30]. Our results indicate that RosA induced more downregulation of IL-4 compared with SXCF, corresponding to higher level of IFN-γ (Figure 4), and IgE was downregulated as low as control group after SXCF treatment (Figure 3E), which means SXCF exhibits greater anti-asthmatic potential.

Remarkable parts of LMs were regulated by both RosA and SXCF in asthmatic mice and PLS-DA were carried out to discover those LMs impacted by modelling. As shown in Figure 6A, 16 LMs of lung tissue were determined to change significantly with OVA sensitization compared with Con group (*p* < 0.05). With SXCF treatment, 50% (8 of 16) of LMs were significantly changed compared with the Mod group, 87.5% (7 of 8) of these LMs overlapped with those significantly changed by RosA (*p* < 0.05); the regulation direction was basically the same. It can be observed in Figure 6C that the effect of SXCF on the metabolism network of LMs was mainly confined to the metabolites of Linoleic acid. RosA has a broader impact; 62.5% (10 of 16) of LMs were significantly regulated by RosA (*p* < 0.05). We determined that only 9,10-DiHOME and 12,13-DiHOME were upregulated by SXCF and RosA compared with most of these downregulated LMs.

As shown in Figure 6B, only a limited fraction of LMs changed significantly in serum after OVA stimulation. In total, 50% (3 of 6) LMs were reversed by SXCF. Only 33% (2 of 6) were regulated by RosA. It was believed that LMs changes in serum would be minimized by dilution effects, as metabolites pool from diverse organs. It should be noted that one LM (5,6-DHET) was reversed, and another LM (8,9-DHET) decreased further by RosA treatment.

### 3.1. RosA Induced Equal Effects on Key-LMs as SXCF

We can observe that most of the LMs show the same trend, such as TXB_2_, 9-HODE, 9,10-DHOME, and 12,13-DHOME (Figure 7). Both RosA and SXCF regulation ratios (R_r_) were mostly distributed from 0.75–1.25 (Table 2).

The most impacted LMs were filtered by *p* value and VIP score. TXB_2_ was the most important LMs after SXCF treatment and RosA treatment (VIP > 1.5). As shown in Figure 4, compared to the Mod group, the TXB_2_ level had a significant reversal upon SXCF administration and RosA administration, even lower than that of the control group.

TXA_2_ is very unstable in aqueous solution; it is hydrated to biological inactivity within about 30 s. In most studies, the TXB_2_ level was extensively examined to reflect the level of TXA_2_. TXA_2_ is a powerful vascular inflammatory factor, and previous research demonstrated that TXA_2_ inhibits dendritic cell-dependent proliferation of T cells and weakens the dendritic cell–T cell adhesion, which proved that RosA and SXCF can reduce inflammation by downregulating TXA_2_ to reach therapeutic effect [31,32]. The release of TXA_2_ from macrophages was stimulated by IgE, and TXA_2_ subsequently stimulated the TNF-α production by human alveolar macrophages [33]. As shown in Figure 4A, TNF-α was downregulated significantly with SXCF and RosA treatment (*p* > 0.05). IgE also has a downregulation in both SXCF group and RosA group. Hayashi et al. reported that TXA_2_ plays an important role in airway allergic inflammation, which is related to the infiltration of Th2 lymphocytes and eosinophils [34]. These findings suggest that TXA_2_ induced IL-4 upregulation. In Figure 6A, TXA_2_ is higher in SXCF group, which is consistent with the increase in IL-4 levels in SXCF group (Figure 4B).

HODEs are generated through the oxidation of linoleic acid and arachidonic acid and have been shown to regulate the inflammation in vascular wall [35]. 13-HODE and 9-HODE are two of the most commonly studied HODEs. They are involved in a variety of physiological processes, including inflammation and angiogenesis [36,37]. The activity of HODEs is regulated by Th2 cytokines, and IL-4 plays an important role in the synthesis of HODEs [38]. As shown in Figure 4B and Figure 6A, HODEs levels are lower in RosA, which means that RosA prevented the formation of HODEs by inhibition of IL-4.

### 3.2. SXCF Reduced the Side Reactions by Neutralizing Parts of RosA Regulation on LMs Metabolic Network

We can also observe the difference between SXCF group and RosA group (Table 2). RosA has excessive regulation on some of these LMs, such as 14,15-DHET, 12,13-DHOME, and 9,10-DHOME (R_r,RosA_ > 125%). These excessive regulations may be the main factor of side reactions of RosA.

DHOMEs are metabolized by linoleic acid through soluble epoxide hydrolase (sEH) and CYP450 epoxygenases [39]. These metabolites induced oxidative stress and chemotaxis of human neutrophils and aggravated lung inflammation in asthmatic mice [40] and also appeared in the airways of adults with asthma following bronchial provocation with an allergen [41]. As reported, intraperitoneal injection of DHOMEs increased pulmonary inflammation manifested as the increase in IgE levels and the decrease in regulatory T cells in the lungs [40]. Here, we showed that overregulated DHOMEs were accompanied by a high level of IgE in RosA groups, which shows the side effect of RosA (Figure 3E and Figure 7).

EETs are known for their anti-inflammatory effect and vascular function [42,43]. 14,15-EET is a lipid mediator produced by the metabolism of arachidonic acid. It is a bioactive molecule that regulates intracellular signaling and physiological processes through interactions with G protein-coupled receptors [44]. 14,15-EET is involved in various physiological and pathological processes, including cardiovascular diseases and inflammation [45]. Studies have shown that 14,15-EET has protective effects on the cardiovascular system, including lowering blood pressure, antiplatelet aggregation, and involvement in vasodilation and angiogenesis [42,43]. Additionally, 14,15-EET has anti-inflammatory and anti-oxidative stress effects, which can alleviate inflammatory reactions and tissue damage caused by oxidative stress [46]. DHETs are the stable metabolites of the EETs, and 14,15-EET is converted to 14,15-DHET almost entirely in six hours [43]. 14,15-DHET is a biologically active molecule that plays a role in regulating various physiological processes, including blood pressure regulation, inflammation, and pain perception. It is also involved in the regulation of vascular tone and endothelial function [47,48]. Studies have shown that 14,15-DHET has anti-inflammatory effects and is involved in the resolution of inflammation [49]. Additionally, it has been suggested that 14,15-DHET may have therapeutic potential in the treatment of various diseases, including hypertension [50] and cardiovascular disease [51]. As shown in Figure 4A, the 14,15-DHET levels increased significantly in the RosA group (*p* < 0.05) and also increased in the SXCF group, which proved the anti-inflammatory effect of SXCF and RosA. Furthermore, 14,15-EET has been shown to have a hypotensive impact [52], while elevated EETs and DHETs levels are associated with ischemic stroke [53]. As shown in Figure 5, 14,15-DHET was overregulated in the RosA group. Excessive secretion of 14,15-DHET may increase the risk of incident ischemic stroke. It is believed some constituents in SXCF besides RosA may neutralize excessive impacts of RosA on LMs.

## 4. Materials and Methods

### 4.1. Materials

MS-grade water, acetonitrile, formic acid, acetic acid, and methanol were purchased from Fisher Scientific (Waltham, MA, USA). Isopropanol was obtained from Honeywell (West Valley City, UT, USA). Glycerol, Ovalbumin (OVA) and Butylated hydroxytoluene (BHT) were purchased from Sigma-Aldrich (Milwaukee, WI, USA). Internal standards were purchased from Cayman (Ann Arbor, MI, USA). IL-4, IL-10, TNF-α, IFN-γ, and IgE ELISA kits were purchased from Lianke (Shaoxing, China). Waters MAX SPE cartridges (Milford, MA, USA) and BCA kit (Rockford, IL, USA) were also obtained.

### 4.2. Methods

#### 4.2.1. Network Pharmacology

We entered the keyword “rosmarinic acid” into PubChem (https://pubchem.ncbi.nlm.nih.gov/, accessed on 6 July 2022) and obtained Isomeric SMILES of RosA.

Isomeric SMILES of RosA was imported into the Swiss Target Prediction website (http://www.swisstargetprediction.ch/, accessed on 6 July 2022) to predict RosA target proteins.

We entered “asthma” in the GeneCards (http://www.genecards.org/, accessed on 6 July 2022) database and DrugBank (https://go.drugbank.com/, accessed on 6 July 2022) database to predict potential asthma target.

The intersection targets of the RosA and asthma were input into the R software 4.1.1(R Foundation for Statistical Computing, Vienna, Austria) data package to perform Gene Ontology (GO) enrichment analysis and Kyoto Encyclopedia of Genes and Genomes (KEGG) pathway analysis on the effective targets of the RosA to explore the biological process and signaling pathway of treating asthma with RosA.

#### 4.2.2. Animals and Preparation of Asthma Model

All the animal experiments were approved by the Animal Care and Use Committee of Xinjiang Medical University, Ethics Committee of Xinjiang Medical University (IACUC-20210603-07). After 3 days of experimental feeding, mice were randomly divided into four groups, Con group, Mod group, SXCF group, RosA group, with five mice per group.

On 1st and 8th day, mice of group Mod, SXCF, and RosA were OVA-sensitized twice by an intraperitoneal injection of 0.2 mL OVA sensitization solution (0.5 mg/mL in aluminum hydroxide gum saline diluent). Con group mice were injected with 0.2 mL saline. From the 15th to 21st day, the Mod group was orally administered with 0.5% CMC-Na, group SXCF and group RosA mice were orally administered with SXCF (10 mg/mL, 100 mg/kg) and RosA solution (0.5 mg/mL, 5 mg/kg) equal to SXCF dose (4.8% *m*/*v* which was measured by UPLC-MS.), respectively. After one hour, group B, C and D were challenged by nebulized OVA (10 mg/mL) for 30 min; Con group mice were challenged by saline. Mice received this treatment once a day and had free access to food and water during the study period.

#### 4.2.3. Hematoxylin and Eosin Staining

For lung histology, lungs were dissected, fixed in 4% paraformaldehyde, paraffin-embedded, sectioned, stained with hematoxylin and eosin, and analyzed under a light microscope.

#### 4.2.4. Cytokine Detection

TNF-α, IFN-γ, IL-4, and IL-10 levels in BALF and serum IgE were measured by specific ELISA kits purchased from Lianke (Shaoxing, China).

#### 4.2.5. Sample Preparation

A total of 50 μL of serum was thawed on ice and 70 μL 10% glycerol aqueous solution was added. Samples were diluted to 1 mL with 25% acetonitrile aqueous solution.

A total of 10 mg of ground lung tissue was mixed with 250 μL acetonitrile. After centrifugation at 15,000 rpm for 5 min, 200 μL supernatant was transferred to a tube containing 70 μL 10% glycerol aqueous solution. Samples were diluted to 1 mL final volume (containing 20 μg BHT and 0.5 ng IS) with 25% acetonitrile aqueous solution.

Solid-phase extraction cartridges were equilibrated with 3 mL acetonitrile and 3 mL 25% acetonitrile aqueous solution. Samples were loaded onto conditioned cartridges. The cartridge was washed with 25% acetonitrile aqueous solution and acetonitrile. LMs were eluted with acetonitrile containing 1% formic acid. The elution was concentrated with a vacuum concentrator. Purified samples were stored at −80 °C before UPLC-MRM analysis.

All samples were reconditioned with 50 μL acetonitrile/methanol (50/50, *v*/*v*) for injection.

The remaining 100 μL of supernatant of lung tissue was mixed with 100 μL of water and applied for protein quantitation by BCA kit.

#### 4.2.6. UPLC-MS/MS Conditions

LC-MS/MS analysis was performed on a AB SCIEX Triple Quad 5500^+^ MS (Framingham, MA, USA) coupled with a Thermo Scientific Dionex Ultimate 3000 UHPLC (Waltham, MA, USA). A Waters ACQUITY UPLC BEH C18 (2.1 × 50 mm) (Milford, MA, USA) was applied. The column temperature was set as 40 °C. Mobile phase A was 0.1% formic acid aqueous solution; mobile phase B was ACN/IPA (9/1 *v*/*v*). Gradient elution conditions are shown in Table 3. The flow rate was 0.4 mL/min. The injection volume was 5 μL. Data acquisition was performed in negative ion mode equipped with an ESI source.

#### 4.2.7. Data Analysis

The data acquisition was carried out with Analyst 1.7.1 (AB SCIEX, Framingham, MA, USA). Integration and quantification were achieved with OS-Q V1.6 (AB SCIEX, Framingham, MA, USA). Principal component analysis was conducted with EZinfo (Waters, Milford, MA, USA). T-tests were performed by IBM SPSS (Armonk, NY, USA). Heatmaps were utilized using the Metaboanalyst 5.0 (http://www.metaboanalyst.ca/, accessed on 30 June 2022).

## 5. Conclusions

According to the results, both SXCF and RosA had the effect of asthma control and inflammation reduction by regulation of LMs associated with cytokine production and leukocyte chemotaxis. This proved that RosA can exert therapeutic functions against asthma as the main constituent of SXCF. LMs overregulation of target organ induced by RosA was neutralized by other constituents in SXCF. There are other substances synergistically enhancing its anti-asthma efficacy in the SXCF, which is better than the equivalent RosA. While further research is needed to understand RosA’s full potential as a treatment for asthma in humans, the existing evidence shows that RosA may be a viable natural product for managing asthma symptoms and inhibiting airway inflammation, and SXCF has the potential to be an asthma treatment option with accessibility, adherence, and safety.

## Figures and Tables

**Figure 1 molecules-28-03827-f001:**
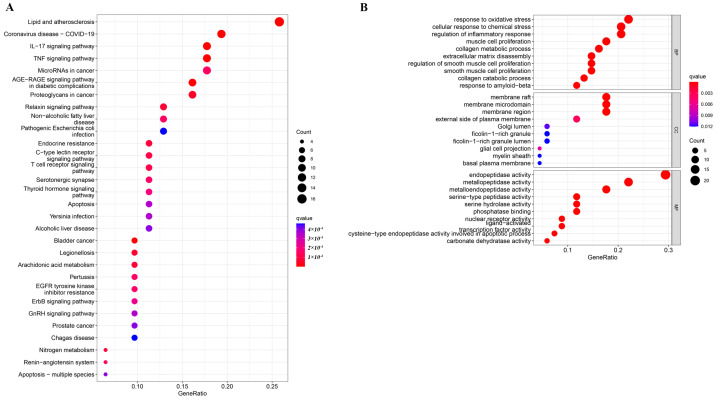
Enrichment analysis of key modules (**A**): Kyoto Encyclopedia of Genes and Genomes (KEGG); (**B**): Gene ontology (GO).

**Figure 2 molecules-28-03827-f002:**
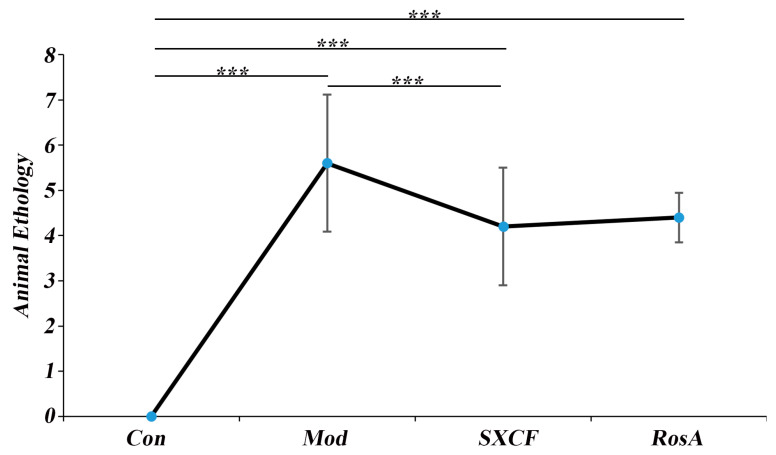
Animal Ethology of Different Groups. ***: *p* < 0.001.

**Figure 3 molecules-28-03827-f003:**
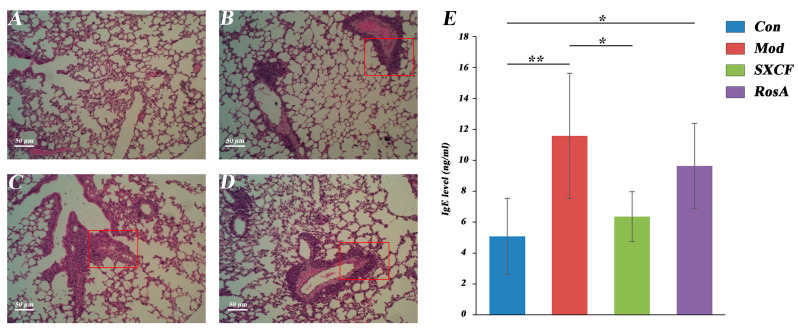
Lung Histology Analysis of Different Groups. (**A**): Con group; (**B**): Mod group; (**C**): SXCF group; (**D**): RosA group; (**E**): IgE content in serum. *: *p* < 0.05, **: *p* < 0.01.

**Figure 4 molecules-28-03827-f004:**
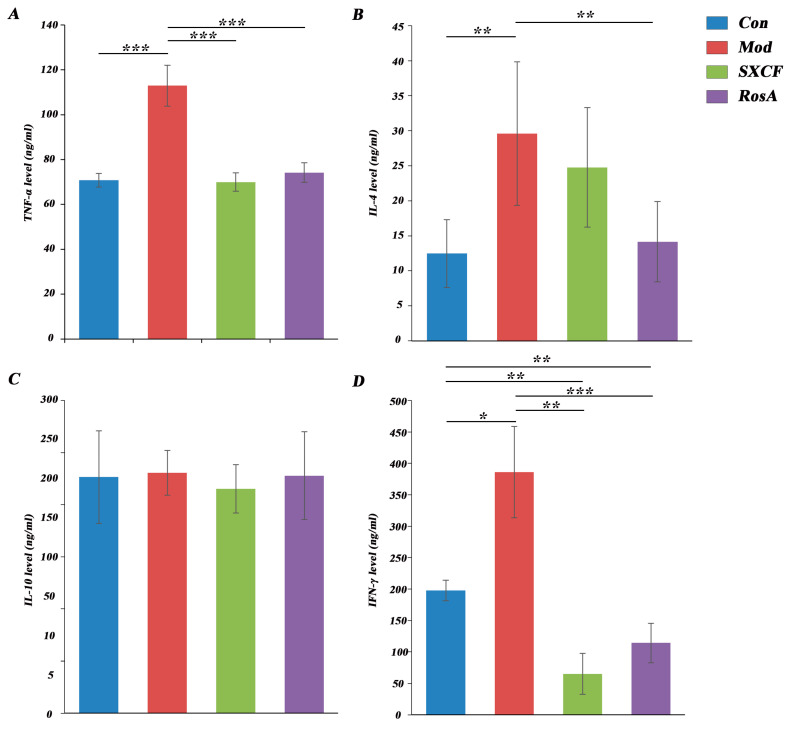
Cytokine content of different groups in BALF. (**A**): TNF-α level; (**B**): IL-4 level; (**C**): IL-10 level; (**D**): IFN-γ level. *: *p* < 0.05, **: *p* < 0.01, and ***: *p* < 0.001.

**Figure 5 molecules-28-03827-f005:**
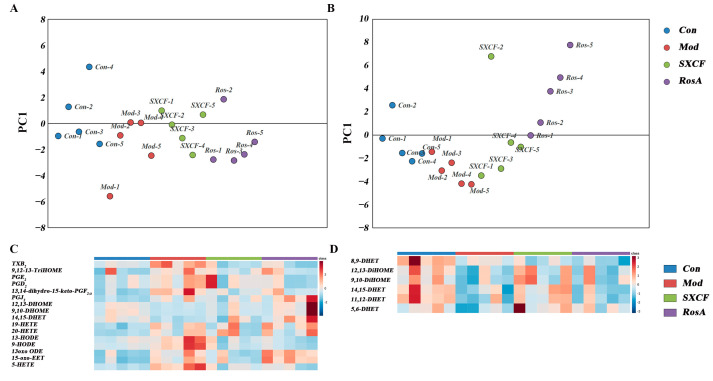
LMs regulation of OVA-induced asthma group and the control group. ((**A**) One-dimensional PCA score scatter plot of lung tissue. (**B**) One-dimensional PCA score scatter plot of serum. (**C**) Heatmap of lung tissue. (**D**) Heatmap of serum).

**Figure 6 molecules-28-03827-f006:**
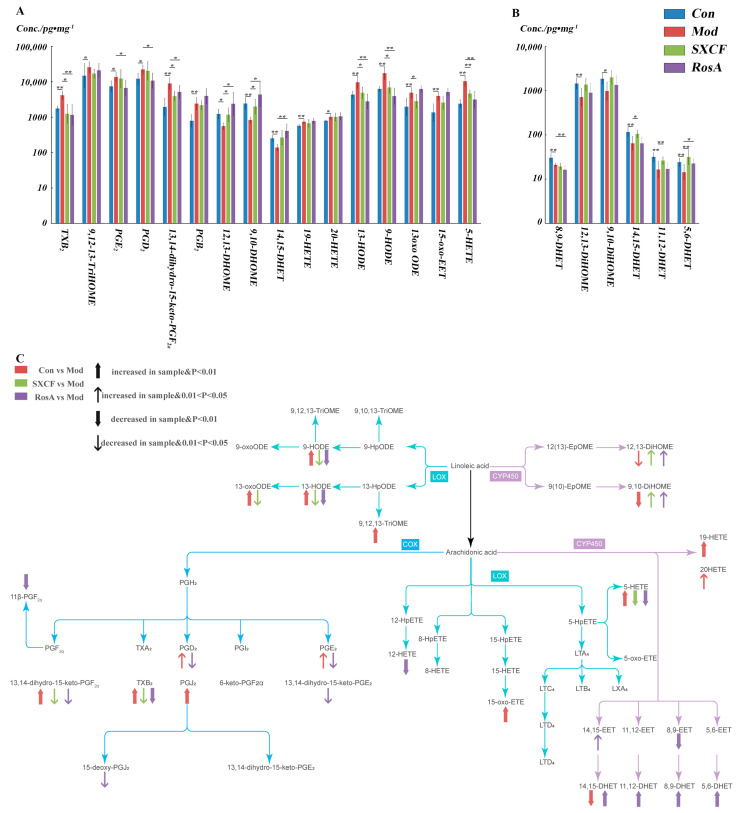
A&B: Different LMs Regulation between SXFC and RosA in target organ (**A**) and serum (**B**). All the LMs shown in the figure were determined to be significantly changed between Con group and Mod group (*p* < 0.05). *: *p* < 0.05 and **: *p* < 0.01. (**C**): LMs detected in target organ with their metabolism pathway and variations among different groups.

**Figure 7 molecules-28-03827-f007:**
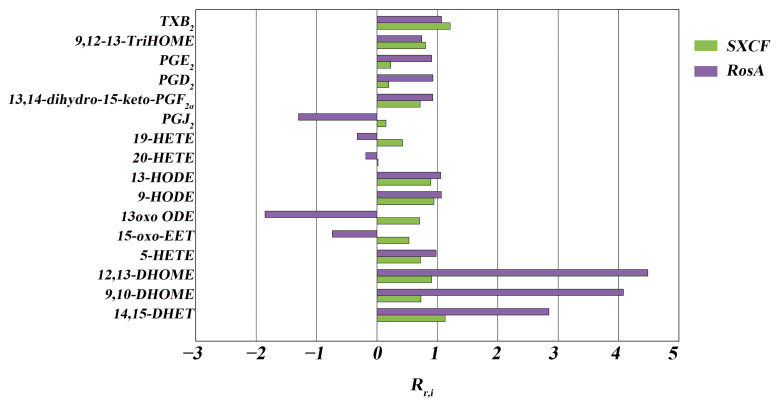
The LMs regulation ratios (R_r_) of SXCF and RosA. LMs shown in the figure were significantly regulated in the Mod group compared with the Con group. $R_{r,i}=\frac{C_{m}-C_{i}}{C_{m}-C_{c}}$, *C_m_*: Concentration of LMs in Mod group; *C_i_*: Concentration of LMs in SXCF or RosA group; *C_c_*: Concentration of LMs in Con group.

**Table 1 molecules-28-03827-t001:** Analysis of mice pulmonary injury.

Group (*n* = 5)	Grading Score for Pulmonary Injury			
	--	+	++	+++
Con	5	0	0	0
Mod	0	1	2	2
SXCF	0	4	1	0
RosA	0	3	2	0

--: The morphology of the lung tissue was normal, and no abnormal changes were seen. +: Mild pathological changes in a small portion of lung tissue (bronchi, pulmonary interstitium). ++: Moderate pathological changes in a portion of lung tissue (bronchi, pulmonary interstitium). +++: Large pathological changes in the majority of lung tissue (bronchi, pulmonary interstitium).

**Table 2 molecules-28-03827-t002:** LMs regulation summary.

Group	*R_r,i_* > 125%	75% < *R_r,i_* < 125%	0 < *R_r,i_* < 75%	*R_r,i_* < 0
SXCF	0	10	6	0
RosA	3	7	1	5

**Table 3 molecules-28-03827-t003:** Gradient Elution program for LMs analysis.

Time/min	A%	B%
0	75	25
1	75	25
8	5	95
8.5	5	95
8.51	75	25
10	75	25

## Data Availability

The data presented in this study are available on request from the corresponding author.

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
