# Peer review of "Lipid Mediators Metabolic Chaos of Asthmatic Mice Reversed by Rosmarinic Acid"

_molecules, 2023, doi:10.3390/molecules28093827_

Round 1

Reviewer 1 Report

Rosemary acid is found not only in Hyssopus cuspidatus, the acid is popular in the Labiatae family. if this species is mentioned, the research was conducted on extracts from this plant? , on which ?. Hyssop also contains other active ingredients. Were the studies conducted on the rosemary acid standard? There are many studies on the anti-asthmatic activity of rosemary acid.

  • DOI: 10.1017/S1466252317000081

doi.org/10.3389/fphar.2023.1066643 

Reviewer 2 Report

The authors state that both SXCF and RosA had the effect of asthma control and inflammation reduction by regulation of LMs associated with cytokine production and leukocyte chemotaxis, in addition, that some constituents in SXCF could neutralize a side effect associated with RosA-induced LMs overregulation. The paper may contain some results of interest, but it has several points that require further attention.

#1 Not showing a significant difference in change of the ethological score and lung histology after OVA sensitizing in SXCF and RosA groups compared with that in Mod group, the therapeutic effect on phenotypes in an animal model of asthma is unclear in this study.

#2 The authors should discuss the implications of the difference between the effect of SXCF and RosA on IgE and IL-4 levels, including the relationship to lipid mediators such as TXA2. Some explanation is required.

Round 2

Reviewer 2 Report

Requests have been fulfilled in present form.